# Research Progress in Distributed Acoustic Sensing Techniques

**DOI:** 10.3390/s22166060

**Published:** 2022-08-13

**Authors:** Ying Shang, Maocheng Sun, Chen Wang, Jian Yang, Yuankai Du, Jichao Yi, Wenan Zhao, Yingying Wang, Yanjie Zhao, Jiasheng Ni

**Affiliations:** 1Laser Institute, Qilu University of Technology (Shandong Academy of Sciences), Jinan 250101, China; 2College of Science, Shandong Jianzhu University, Jinan 250101, China

**Keywords:** distributed acoustic sensing, optical fiber sensor, optical time domain reflectometry, Rayleigh backscattering, performance boost

## Abstract

Distributed acoustic sensing techniques based on Rayleigh scattering have been widely used in many applications due to their unique advantages, such as long-distance detection, high spatial resolution, and wide sensing bandwidth. In this paper, we provide a review of the recent advancements in distributed acoustic sensing techniques. The research progress and operation principles are systematically reviewed. The pivotal technologies and solutions applied to distributed acoustic sensing are introduced in terms of polarization fading, coherent fading, spatial resolution, frequency response, signal-to-noise ratio, and sensing distance. The applications of the distributed acoustic sensing are covered, including perimeter security, earthquake monitoring, energy exploration, underwater positioning, and railway monitoring. The potential developments of the distributed acoustic sensing techniques are also discussed.

## 1. Introduction

Optical fiber sensing techniques are an important means of evaluating the degree of a country’s informatization [1,2]. Scattered light in the optical fiber is used as the information carrier to sense and transmit the changes in external physical quantities. The scattered light in an optical fiber includes Raman, Brillouin, and Rayleigh scattering. Among these, the first two types are related to the vibrationally excited state of the optical fiber, and both involve inelastic scattering. The difference between the two types is that the former interacts with optical phonons and the latter interacts with acoustic phonons [3,4,5].

Rayleigh scattering was introduced because the inhomogeneous refractive index is generated by the inhomogeneous distribution of the optical fiber material [6]. Rayleigh scattering is a linear process because the scattered power is proportional to the incident power. In addition, it is also known as elastic scattering because the frequency of the scattered light does not change compared to the incident light. To date, it has been widely used in the field of distributed sensing because of its strong scattered light intensity and lack of frequency shifting. Thus, distributed acoustic sensing (DAS) techniques based on Rayleigh scattering have attracted intensive research due to their unique distributed sensing performance and high sensitivity measurement capability in applications [7,8,9,10,11].

Distributed detection is used to measure the variation information along the sensing fiber by detecting the backscattering. In 1976, Barnoski et al. first proposed the optical time domain reflectometer (OTDR) technique according to the design concept of LiDAR [12] and applied it to detect the loss of optical fiber links.

Since the OTDR cannot respond to the phase modulation information caused by interference events, Healy et al. proposed the concept of coherent OTDR (COTDR) in 1982 to further improve the performance of the system [13]. In 1993, Taylor et al. proposed a high-sensitivity phase-sensitive OTDR (Φ-OTDR) technique [14], from which DAS entered the qualitative detection stage.

In order to extract the information of external physical quantities, the phase demodulation technologies are proposed to demodulate the interferometric signal within the RBS. In 2013, Newson et al. proposed phase demodulation based on a 3 × 3 coupler [15]. In 2015, Li et al. proposed a phase-generating carrier (PGC) demodulation method [16]. In 2016, Rao et al. used IQ demodulation to process optical fiber stretching signals [17], from which DAS entered the quantitative detection stage.

Currently, DAS techniques have undergone great development due to the improvement in performance indicators [18] (Figure 1), such as polarization fading, coherent fading, spatial resolution, frequency response, signal-to-noise ratio, and detection distance.

In this paper, first, the basic sensing principles of the DAS system are introduced, and then the technical difficulties and solutions of DAS techniques in terms of polarization fading, coherent fading, spatial resolution, frequency response, signal-to-noise ratio, and detection distance are demonstrated. The latest progress of DAS techniques is also described in the fields of perimeter security, earthquake monitoring, energy exploration, underwater positioning, and railway monitoring. Finally, a summary of DAS techniques is presented.

## 2. Basic Sensing Principle

OTDR is the basis for distributed detection. DAS systems mainly include the phase sensitive optical time domain reflectometer (Φ-OTDR) and the coherent optical time domain reflectometer (COTDR). When a certain length of the sensing optical fiber is immersed in the external physical field environment (such as acoustic wave, temperature, vibration, or strain), the unit length and refractive index of the optical fiber are changed via the elasto-optical or thermo-optical effect, which causes the optical features (amplitude or phase) of the RBS within that optical fiber. The quantity of the external physical field can be recovered by detection and demodulation.

### 2.1. Principle of OTDR Techniques

When an optical pulse is injected into the optical fiber under test (FUT), the RBS generates different round-trip times at different positions, which are received by the photodetector (PD). The RBS intensity at each position of the optical fiber is obtained by analyzing the electrical signal output from the PD. The position of the scattering point is related to the return time of the RBS. The round-trip time at the input can be expressed as:(1)$$z=\frac{c\sigma }{2n}=\frac{v_{g}\sigma }{2},$$
where *σ* is the time of backscattered detection, *v_g_* is the group speed in the optical fiber, and the factor of 2 means that only the backscattered pulse is needed to return to the detector. Under ideal conditions of the uniform refractive index of the optical fiber, the amplitude of the backscattered light at any point is proportional to the amplitude of the forward propagating light at that position, because Rayleigh backscattered light is a linear process [19] (Figure 2).

### 2.2. Principle of Φ-OTDR Techniques

To address the defect that traditional OTDR techniques cannot respond to external interference, Φ-OTDR techniques evolved from traditional OTDR techniques; the main difference between the two is the choice of laser, with Φ-OTDR techniques having more demanding requirements. In Φ-OTDR, the line width of the laser is very narrow, usually less than 100 kHz, so the coherent length is much longer than the pulse width. When a certain section of optical fiber is affected by interference, it changes the phase of the RBS passing through the corresponding position, leading to the change in the RBS’s intensity due to the interference effect.

A narrow linewidth laser (NLL) emitting highly coherent light is used as the light source, an acousto-optic modulator (AOM) is used to convert the continuous light into a probe pulse, and an erbium-doped fiber amplifier (EDFA) is used to compensate for the previous optical path and the loss of power to the optical path devices. The amplified detection pulses are injected into the sensing optical fiber via a circulator (CIR). Then, its RBS light is delivered through the CIR to a PD [20] (Figure 3).

### 2.3. Principle of COTDR Techniques

Coherent detection techniques were introduced on the basis of Φ-OTDR; they process the beat frequency signal of the local signal and the Rayleigh backscattering signal, and can sense the phase information and position of the external vibration signal in real time. The main difference between COTDR and traditional OTDR is that the former uses a narrow linewidth laser having a stable frequency and much longer coherence length than that of a FUT. By comparing the electrical fields before and after modulation, it can be expressed as:(2)$$E_{l}\left(t\right)=\sqrt{P_{l}}e^{j2\pi v_{0}t},$$
and:(3)$$E_{p}\left(t\right)=\sqrt{P_{p}}e^{j(2\pi v_{0}t+2\pi f_{0}t)}\alpha \left(\frac{t}{H_{p}}\right),$$
where *α*(*t*) is the window function, *H_p_* is the probe pulse width, *f*_0_ is the carrier frequency, *P_p_* and *P_l_* are the power of the RBS light and the local light respectively, and *v*_0_ is the laser’s center frequency. Then the photoelectric field of the PD can be expressed as:(4)$$E_{R}\left(t\right)=\sum_{i=1}^{N}E_{i}e^{j[2\pi v_{0}(t-\sigma_{i})+2\pi f_{0}(t-\sigma_{i})]}\alpha \left(\frac{t-\sigma_{i}}{H_{p}}\right),$$
where *N* is the number of the scattering points along the FUT, *σ_i_* is the round-trip time of the *i* scattering point, and *Ei* is its photoelectrical field amplitude [21] (Figure 4).

## 3. Research Progress

### 3.1. Polarization Fading

The birefringence phenomenon results in polarization fading because the polarization state of the output light is different from that of the reference light [22]. The main feature of polarization fading is the random fluctuations of amplitude of the beat frequency signal of the interference between the signal light and the reference light. The amplitude of the scattered waveform is close to zero at some positions when polarization fading occurs in the phase demodulation.

As early as the 1970s, the fading phenomenon was preliminarily studied in the fields of optical imaging [23] and wireless communication [24]. At present, it is addressed using polarization-maintaining fiber depolarization [25], Faraday rotating mirrors [26], input polarization state control [27], high-speed polarization modulation [28], diversity reception [29], etc.

In 2014, Wu et al. proposed a stable coherent and polarization maintaining light path structure with up to 40% visibility of interference fringes [30]. With the development of the technique, dual frequency probe pulses could be used to suppress polarization fading. In 2015, Alekseev et al. used the dual-pulse diverse frequency probe signal for phase signal reconstruction at any positions on the Φ-OTDR system, and experimental results demonstrated the feasibility of the scheme [31]. Although the scheme had significant limitations and was not yet mature enough for the control of both pulses, the experiment demonstrated the feasibility of the method and provided a direction for later researchers to suppress polarization fading. In 2017, Chen et al. proposed a new form of phase detection that effectively suppressed the effect of polarization fading. Experiments showed two simultaneous vibrations were detected in a 35 km optical fiber with an SNR of more than 26 dB [32].

In 2020, Sun et al. proposed a distributed optical fiber acoustic sensing demodulation scheme based on a dynamic birefringence estimation [33] (Figure 5). The experimental results showed the effective suppression of the drastic polarization change, which was about 9.5 dB. The uniform background noise averaged about 1.2 × 10^−3^
$\mathrm{Rad}/\sqrt{\mathrm{Hz}}$ at different positions.

In 2020, Rao et al. first proposed the bipolar Golay coding Φ-OTDR laser scan-rate problem with heterodyne detection function and adopted a real-time compensation (frequency drift compensation for the laser) method for its solution [34]. The method combined the spectrum extraction and remix methods to suppress polarization fading. Experiments showed that, compared with the unipolar code case, the SNR was improved by 7.1 dB within the sensing range of 10 km, the spatial resolution reached 0.92 m, and the measurement time was 1/2 of the original. By solving the frequency drift and fading problems, the distributed sensing capability of optical pulse coding (OPC) Φ-OTDR was realized, and the bipolar scheme could be applied in many other coding schemes, providing more possibilities for OPC to enter Φ-OTDR.

In 2020, Guerrier et al. proposed a coherent-MIMO sensing technique, which was based on the Φ-OTDR system [35] (Figure 6). The transmitter adopted double polarization multiplexing and the receiver adopted polarization diversity. A comparison of two-phase estimation methods for multiple polarization input-multiple polarization output sensing (MIMO) and single polarization input–multiple polarization output sensing (SIMO) led to the conclusion that coherent-MIMO sensing techniques outperform partial polarization diversity sensing techniques in terms of sensitivity. The polarization effect had little influence on the double polarization detection of the optical fiber sensor, which reduced the probability of false alarms in the system and greatly improved its sensitivity. It is of great significance to further study the double polarization demodulation of the optical fiber sensor.

In 2021, Gu et al. proposed a new spatial diversity technique based on multi-core optical fiber [36] (Figure 7). By means of a fan-in and fan-out module, the independent transmission and centralized reception of signals were carried out in the four cores of the multi-core optical fiber, and the multiple signals were effectively combined by the coherent combining techniques. The experimental results showed that the external interference signal was well reconstructed, the signal fading was suppressed, and the noise floor of the system was decreased to 5.2 dB compared with the optimized single-mode optical fiber system. The sensor exhibited a high level of performance with a minimum noise floor of −85 dB.

In 2021, Ogden et al. analyzed a COTDR system that was based on a frequency multiplexed pulse sequence structure [37]. The method was implemented by increasing the average power injected into the optical fiber, thereby suppressing polarization fading while reducing noise, improving the linearity of the sensor, and achieving a minimum detectable strain of 0.6 $\;p\varepsilon /\sqrt{\mathrm{Hz}}$. The research progress for suppressing polarization fading is summarized in Table 1.

In summary, to solve the polarization fading problem, researchers changed the optical path structure at an early stage. However, the suppression effect of polarization fading problem was not obvious due to the hardware problem at that time. In recent years, phase extraction was proposed by using the dual-pulse diverse frequency probe signal. This is able to detect two simultaneous vibrations, but still has limitations and cannot be carried out on a large scale. Recently, researchers started from the principle of the generation of polarization fading, and the method of simultaneous improvement of software and hardware, such as the demodulation scheme based on dynamic birefringence, was demonstrated. Frequency drift was proposed using the bipolar Golay coding technique with the heterodyne detection function. This significantly improved the system performance compared with traditional unipolar coding, but spatial resolution must be improved. The sensitivity was optimized by spatial diversity techniques, but the experimental system was more complex. The changing pulse sequence technique was used for fading suppression and can greatly reduce the noise, but it requires a large amount of technical support and the system cost is high. In general, the bipolar coding technique, spatial diversity technique, and changing pulse sequence technique are not mature enough, but this does not affect their future improvements for solving the problem of polarization fading and improving the system performance.

### 3.2. Coherent Fading

Coherent fading relates to light fluctuating up and down of Φ-OTDRs when narrow linewidth lasers having a long coherence length detect intra-pulse interference of the RBS generated by the pulsed light. The RBS become weaker or even converges to zero at certain locations, resulting in random detection blind spots in the time and frequency domains because of coherent fading. In turn, the process of phase demodulation causes a sharp deterioration in the signal-to-noise ratio, and the reconstructed phase information of the external signal is far from the actual situation, leading to serious consequences of missed or even false alarms [38,39].

In order to solve the problems caused by coherent fading, researchers conducted a series of studies, such as pulse coding techniques, frequency division multiplexing (FDM), and internal pulse division methods [40,41]. The pulse coding techniques using the multi-frequency pulse method have poor fading suppression because they need different detection pulses of different frequencies, and exhibit a large variability from pulse to pulse. Based on the above problems, in 2018, Cai et al. proposed a new method based on the differential phase shift pulse (DPSP) technique to improve the original phase extraction method. The phase can be demodulated by the amplitude threshold to reduce the probability of coherent fading [42].

In 2021, He et al. proposed a phase-shift transform method to suppress the coherent fading of Φ-OTDR based on the original multi-frequency pulse method. The detected signal was first decomposed and the π phase shift of a signal with complementary amplitude was obtained. The false phase was corrected by synthesizing the complementary signal. This experiment not only allowed the intensity fluctuations above 60 dB to be reduced to 15 dB, but also reduced the standard deviation of the differential phase to 0.0224 [43].

To ensure the desired outcome without sacrificing spatial resolution, in 2019, Zhang et al. proposed a coherent fading suppression method based on frequency division multiplexing (FDM) Φ-OTDR to keep the signal distortion induced by coherent fading in the order of 10^−2^ [44] (Figure 8).

In order to meet the performance requirements of commercial fields such as wireless communication, FDM based on the above proof can effectively suppress coherent fading. In 2021, Zhang et al. proposed a method to suppress coherent fading by Φ-OTDR based on space division multiplexing (SDM), and the experimental results proved that the method could greatly reduce the distortion rate of the signal to maintain it below 2%. This was useful in significantly improving the monitoring performance of commercial systems [45].

In 2021, He et al., proposed a method using time-gated digital-optical frequency domain reflectometry (TGD-OFDR) to suppress the coherent fading of Φ-OTDR [46] (Figure 9). The chirped pulses were divided into overlapping bands and reassembled after digital decoding to achieve a maximum detectable range of 80 km. The superiority of the approach was proven in practical tests. The research progress for the suppression of coherent fading is summarized in Table 2.

In summary, a large number of techniques have been proposed to suppress coherent fading. The DPSP can reduce the probability of interference without sacrificing the vibration response bandwidth, but cannot be widely used due to the limitation of complex experimental operations and performance loss. The phase shift transform technique greatly improves on the weaknesses of the traditional multi-frequency pulse method and has the ability to correct for shifted phases; moreover, although the system is more complex, it does not require complex frequency/phase modulation. FDM can not only suppress the distortion rate of the signal caused by coherent fading to 1.26%, but also improves the overall frequency corresponding range, which requires greater hardware and changes the original system structure. The SDM technique is not as accurate as the FDM technique, but its structure is simple and it can effectively be adapted to commercial use. The TGD-OFDR technique can ensure scattered light detection with a high enough SNR, and its commercial performance index is also relatively outstanding, exceeding other commercial devices in traditional SM fiber.

### 3.3. Spatial Resolution

Spatial resolution is the minimum distance that an optical fiber sensing system can effectively identify two individual events. It is one of the main parameters used to measure DAS system performance.

In general, the spatial resolution is mainly influenced by the pulse width, which can be written as:(5)$$\Delta z=\frac{cT_{w}}{2n},$$
where Δz is the spatial resolution, c is the speed of light, and $T_{w}$ and n are the pulse width and the refractive index of the optical fiber, respectively. However, the pulse width is inversely proportional to the SNR and the sensing distance. Determining how to balance these three parameters is of great significance to researchers.

Improved sensing system structure and optical path devices have been used to improve the spatial resolution of the system. In 2016, Shang et al. added an interferometer on the basis of the traditional optical path, recovered the phase information via the phase carrier demodulation algorithm, and realized a flat frequency response curve and 10 m spatial resolution [47]. In response to the shortcomings of conventional piezoelectric transducer (PZT) modulation, such as low efficiency and insufficient performance, in 2021, Ma et al. proposed an optical fiber PGC modulation structure based on a LiNbO_3_ through-waveguide phase modulator [48]. This structure had greatly improved performance compared to the traditional interferometer, was capable of detecting weak acoustic signals, and achieved a spatial resolution of 10 m. It provided a new research idea for the development of DAS systems. In 2021, Zhu et al. established a new Φ-OTDR optical path system [49] (Figure 10). This system used a distributed feedback (DFB) semiconductor laser combined with an optical waveguide ring resonator (OWRR) as the light source, and its linewidth and stability were excellent at a reduced cost. It offered the advantages of compactness, ease of integration, and high interference immunity. Simultaneous measurements of two vibration sources could be made over 4700 m of fiber with a spatial resolution of 13 m.

High spatial resolution was achieved using narrow width optical pulses or by introducing a swept pulse compression mechanism. In 2019, He et al. proposed a distributed acoustic sensor scheme that was independent of polarization fading that overcame the trade-off between spatial resolution and the sensing distance of conventional Φ-OTDR [50]. The spatial resolution was determined by the bandwidth and mismatch ratio of the chirped pulses rather than the pulse duration, so the spatial resolution could be adjusted to suit the actual requirements by varying the mismatch ratio. The system was available with a spatial resolution of up to 2 m. In 2021, Wang et al. proposed a new method for birefringence measurement using the RBS wave in a single-mode optical fiber [51] (Figure 11). The experiment showed a spatial resolution of 8.6 cm and an average birefringence of 0.234 rad/m. It was shown for the first time that spatial resolution was essential for optical fiber birefringence measurement, and provided an effective tool for characterizing the polarization properties of optical fiber links. In 2021, Qian et al. proposed the chirped pulse conversion algorithm (CPCA), which was based on converting a normal detection pulse into an equivalent chirped detection pulse by convolving the chirp coefficients of the received signal from a Φ-OTDR system [52]. The algorithm demodulated the chirped pulse Φ-OTDR in the Rayleigh interferogram pattern (RIP) to quantify the dynamic strain of the conventional Φ-OTDR. In contrast to the complex and expensive drawbacks of conventional chirp modulation, the generation of equivalent chirp pulses by means of digital processing had the advantage of being simple and inexpensive. The method allowed full quantification of the perturbation to achieve a spatial resolution of 4 m.

To determine the relationship between spatial resolution and SNR, in 2019, Zhang et al. proposed a sensing scheme with multiple spatial resolutions (MSRs) for analyzing Φ-OTDR sensing systems. This scheme could recover vibration events with different interference ranges in a single test with the best SNR while maintaining the same detection frequency range. The results demonstrated that it was extremely important to select the proper spatial resolution, which was beneficial to improving the SNR of the sensing system [53].

Multiplexing techniques can also improve the spatial resolution of the sensing system. In 2021, Gong et al. proposed an OTDR system for dense wavelength division multiplexing passive optical networks (DWDM-PONs). The system was selected to achieve wavelength tunability by selecting an integrated tunable laser assembly (ITLA) as the light source and using wavelet denoising to reconstruct the pulsed light to achieve a spatial resolution of 2 m [54] (Figure 12). The research progress for spatial resolution enhancement is summarized in Table 3.

In summary, the previous researchers added an interferometer from the optical path structure to improve the spatial resolution of the system, but the system structure was complex. Recently, they complemented the interferometer with a laser structure, such as the LiNbO_3_ straight-through waveguide phase modulator and DFB lasers, which increased the system cost but provided more hardware options to improve the spatial resolution. In terms of techniques, pulse compression techniques, multiplexing techniques, and distributed optical amplification techniques have become the main trend in development because they cannot only greatly improve the spatial resolution of the sensing system, but also optimize the optical path.

### 3.4. Frequency Response

In the DAS system, the frequency response reflects the characterization of the frequency range of the system response to external disturbances. The higher the frequency response, the wider the application range of the system. Moreover, more kinds of signals can be effectively detected. However, the time intervals between the detection of optical pulses cannot be less than the round-trip time of the light in the optical fiber, so the frequency response of the system is limited by the sensing range, and these factors are inversely proportional to each other. Determination of how to balance the relationship between these two actors has become an important part of the development of DAS techniques.

In order to balance the relationship between these two factors, researchers carried out a large amount of research in recent years. In 2016, Li et al. proposed a broadband double-frequency ultrasound measurement system for distributed fiber laser sensors in liquid media [55]. In comparing various fiber laser sensors, this system proved that DBR optical fiber laser sensors performed better in broadband double-frequency ultrasound measurements. In 2018, Shang et al. proposed a Φ-OTDR system using broadband weak optical fiber Bragg grating arrays to achieve large temperature resistance of the distributed acoustic sensor [56]. Simultaneous tests at 18 and 50 °C with large local temperature differences resulted in a relatively flat frequency response from 20 to 1200 Hz. In 2021, Yan et al. proposed an ultra-long distributed sensor based on forwarding transmission, coherent detection, and frequency-shifted optical delay lines for ultra-wide frequency from infrasound to ultrasound testing [57] (Figure 13). Compared with the existing distributed sensors, this scheme had the advantages of simple system and sensing structures, ultra-wide frequency response, and ultra-long sensing distance. It enabled ultra-long distributed sensing and could be used to greatly improve performance indicators of the DAS systems.

In addition to improving the system structure, the researchers also adopted the idea of frequency division multiplexing to process the collected signals to realize the spread spectrum of the Φ-OTDR system. In 2019, Zhang et al. proposed a Φ-OTDR system based on an ultra-weak optical fiber Bragg grating (UWFBG) array and frequency division multiplexing (FDM) scheme to expand the frequency response bandwidth (FRB) of the Φ-OTDR system [58]. The experimental results showed that vibration frequencies up to 440 kHz could be detected along the 330 m UWFBG. This was about 3 times higher than the upper FRB limit of conventional systems, and provided a wider FRB and enhanced visibility characteristics for the performance enhancement of the Φ-OTDR system. In 2021, a quasi-DAS system based on a heterogeneous frequency double pulse chain and an array of WFBGs was proposed by Liu et al. [59]. Interference signals at different carrier frequencies were obtained by injecting four different sets of double pulses continuously into a weak optical fiber Bragg grating (WFBG)-sensing optical fiber. This achieved a detection frequency response of 2 kHz and provided a direction for the development of high response frequency for the DAS systems. In 2021, He et al. proposed a new type of distributed optical fiber acoustic sensor based on time delay sampling and frequency division multiplexing of sparse-wideband signals [60] (Figure 14). The sensor could detect two vibration frequencies at the same position, and, by colliding with the frequencies of these two vibrations in three sampling sequences, demodulation could be performed. The system achieved a high SNR of 25 dB, and addressed the trade-off between the measurable distance and the maximum measurable frequency.

The above studies all involved single-mode fibers; however, as the use of multimode fibers has increased, improving their frequency response is urgently needed. In 2021, Murrey et al. proposed a distributed multimode optical fiber Φ-OTDR sensing system [61]. A high-speed camera was used to collect the Rayleigh backscattered light and build a complete backscattered speckle field together with a local oscillator. It achieved a bandwidth of 400 Hz over 2 km of multimode optical fiber. The research progress for spatial resolution enhancement is summarized in Table 4.

In summary, the researchers previously improved DAS systems using hardware parts as lasers and advanced the maturity of the DAS system structure. Recently, the UWFBG method, the FDM method, and a combination of the two methods has been used to enhance the frequency response of the sensors, providing a significant improvement compared to traditional methods. However, the UWFBG method requires complex structure and has high costs. The FDM method requires more sophisticated demodulation algorithms, and was selected as the most suitable method after careful consideration by researchers.

### 3.5. Signal-to-Noise Ratio

The signal-to-noise ratio (SNR) is an important index of the DAS system. The greater the noise, the worse the quality of the obtained signal, and the lower the SNR. This leads to the insensitivity of the system to the signal, reducing the overall performance of the DAS system, and having serious consequences such as missed or even false alarms. The main sources of DAS noise include environmental noise, fading noise, mode noise, and system noise, of which mode noise is affected by polarization fading, coherence fading, fiber strain, and nonlinear effects.

In order to improve the SNR of DAS systems and enhance their sensitivity, researchers have undertaken several studies. In the previous sections, the SNR was improved by reducing the noise. However, the improvement can also be achieved by compensating for or reducing the transmission loss. In 2017, Zhang et al. used optical fibers embedded with UWFBGs for dynamic strain measurement of Φ-OTDR and demodulated the signal phase by an asymmetric 3 × 3 coupler [62]. Experimental results demonstrated that the system could obtain an SNR higher than 56 dB. In 2020, Yang et al. proposed an enhanced distributed optical fiber sensor based on UWFBGs to improve the system SNR, which achieved a system SNR higher than 59.2 dB by using an unbalanced Michelson interferometer (MI) and a 3 × 3 coupler for phase modulation [63]. Based on this, in 2022, Yang et al. used UWFBG and coherent detection to demonstrate that a high extinction ratio and balanced input pulse optical power could improve the performance of the sensing system to obtain a higher SNR [64] (Figure 15). The system structure can also be changed to improve the system SNR. In 2021, Cai et al. proposed a dense multichannel integrated DAS system to solve the system noise problem while eliminating the fading problem, and experimentally demonstrated that the method improved the system SNR by 20 dB [65].

In 2020, Jin et al. first adopted an acousto-optic modulator cascaded with a semiconductor optical amplifier to improve the extinction ratio of the system, and later used time-frequency analysis and minimum mean square error algorithms for amplitude demodulation and phase demodulation to improve the SNR of the system to 42.2 dB [66] (Figure 16). Due to their popularity, artificial intelligence (AI) and machine learning (ML) can be applied to DAS systems to improve the system SNR. In 2021, Zhang et al. proposed a method using the optimal peak-seeking algorithm combined with machine learning for signal identification, which greatly improved the system SNR, and the experimental results provided potential applications for Φ-OTDR devices and future implementations of machine learning algorithms [67]. The research progress for SNR enhancement is summarized in Table 5.

In summary, although special fibers can effectively improve the SNR of the system, it also has the disadvantage of high cost and increasing the complexity of the system. In recent years, researchers also adopted algorithms to improve the SNR of the system; for example, the least mean square error algorithm has the advantages of low cost and simple operation. The introduction of artificial intelligence and machine learning models has greatly improved the performance of the system by processing the signals. This may become a popular direction in the future, but the operation is relatively difficult and requires more advanced technical conditions.

### 3.6. Detection Distance

Optical fiber is a sensor of the DAS system. Under ideal conditions, the transmission of optical fiber is loss-free, but in the real state, the loss of light will increase with the increase in the transmission distance. The detection distance is proportional to the amount of detected light energy, and the main method of increasing the detection distance is to increase the energy of the detected light. Researchers first used optical amplifiers to increase the optical power of incident light, but was not able to amplify without limit, and was limited by nonlinear effects.

In order to perform long-distance detection and eliminate the drawbacks of traditional long-distance detection methods, in 2018, Fu et al. designed a hybrid DAS system integrating a Brillouin optical time domain analyzer (BOTDA) and Φ-OTDR with a sensing distance of 150.62 km [68]. However, in the course of the study, it was found that the nonlinear effects in the stimulated Brillouin scattering had a more serious impact on the detection distance compared with the stimulated Raman scattering. In 2019, He et al. proposed a long-distance and high-sensitivity DAS system based on the time-gated digital optical frequency domain reflection method, which used bidirectional distributed Raman amplification to achieve long-distance measurement [69] (Figure 17). The length of the experimental optical fiber was about 108 km. For the first time, a strain sensitivity of 220 and harmonic-free linear inversion were achieved on a 100 km optical fiber.

In 2019, Cedilnik et al. proposed a maximum reachable DAS without inline amplification [70]. Up to 112 km could be achieved without any optimization, extending the coverage of any DAS system by optimizing the form of optical fiber combinations. This DAS system also has an extended range using a single standard optical fiber. The creation of these two methods will enable future DAS systems to move over long distances.

In addition to the widely used distributed amplification method, in recent years, researchers have used low-loss enhancement fibers to enhance the detection distance. In 2019, Uyar et al. proposed an ultra-long range distributed optical fiber acoustic sensing system using a double acoustic light modulator and a double photodetector technique [71] (Figure 18). The double acoustic optical modulator scheme reduced the coherent noise by generating optical pulses with an extinction ratio of less than 110 dB, while the double photodetector scheme was designed to achieve a high dynamic range. The system was selected to process a signal of 102.7 km, yielding the maximum SNR of 24.7 dB. This was the highest distance reported for a Φ-OTDR distributed acoustic sensor system based on direct detection.

In 2021, Masoudi et al. proposed a DAS with a sensing range of more than 150 km by adding a low-loss enhanced backscattering optical fiber to a single-mode optical fiber [72]. The measurement system had a frequency range of 0.1 to 100 Hz and a spatial resolution of 5 m. The minimum detectable strain at 1 Hz for this combined system was 40 nε. The research progress for detection distance enhancement is summarized in Table 6.

In summary, two optical path systems can greatly improve the detection distance. However, researchers must consider how to maximize the performance of these systems. Improving the hardware facilities of the optical path system, such as with the use of bi-directional distributed Raman amplification or two cascaded acoustic-optical modulators, can improve detection distance, but increases the complexity of the system while increasing the operational difficulties. The low-loss optical fibers have the disadvantage of high cost, and require the researchers to carefully consider and select the most suitable method.

## 4. Application

### 4.1. Perimeter Security

Perimeter security has long been a core condition for the safety of people’s lives and property, and national political stability. It plays an important role in border lines, railway stations, airports, gas stations, large substations, and other areas [73]. The DAS system has the characteristics of a wide monitoring range, a high degree of concealment, strong environmental adaptability, and lack of a blind area. It is highly suitable for application in the field of perimeter security. In recent years, the perimeter security research related to DAS has continued to develop, and the challenge for perimeter security projects is to improve the classification and recognition effect.

Conventional class recognition algorithms have low accuracy. Although deep learning-based classification and recognition algorithms have high accuracy, they take a long time to train and require a large amount of computation. In 2021, Shi et al., from Shantou University, proposed an event recognition method based on transfer training. The experiment was conducted on 4252 groups of samples based on 8 events; Alex Net was pre-trained for 1/5 of the samples, and then trained for the remaining samples. Partial training achieved a classification accuracy of 96.16% in less than 5 min [74] (Figure 19).

In field applications, multiple vibrations often occur in close positions, resulting in the collected vibration signals being mixed signals of multiple signals. To improve the accuracy of intrusion classification at a reduced cost, in 2002 Ni et al., from the Laser Institute of Shandong Academy of Sciences, proposed a recognition algorithm, 100 G-Net, based on a group convolution neural network. The recognition of nine common signals including four mixed signals was realized. Under the condition of a recognition speed of 20 ms/sample, the recognition accuracy of the verification set reached 97.5% [75] (Figure 20).

Existing optical fiber sensing technologies and data analysis methods have been combined to reduce system complexity. In 2021, Shi et al. proposed an interferometric optical fiber perimeter security system that was based on multi-domain feature fusion and support vector machines (SVMs). The system was used to classify and identify non-intrusion, climbing, shaking, iron bar knocking, and optical fiber cable shearing, and achieved an average classification accuracy of 94.4% [76] (Figure 21). The latest progress for perimeter security research is summarized in Table 7.

### 4.2. Earthquake Monitoring

Earthquakes are among the disasters that endanger people’s lives and the safety of property. Research into earthquake monitoring is important to ensure people’s safety and social stability. The conventional earthquake monitoring method requires dense placement of earthquake monitoring instruments on the ground surface, excavation, and backfilling of the involved strata when laying sensing optical fiber cables, which greatly increases the project cycle and cost. By comparison, the DAS techniques realizes the dynamic strain detection of the optical fiber by measuring the phase change of the backscattered light in the optical fiber and then realizes the recording of the earthquake wave field [77]. This is expected to solve the current problems of the high data acquisition cost, limited coverage, and unsuitability for urban implementation in seismic detection [78].

In December 2018 and December 2019, Wang et al. conducted observation experiments twice in the urban area of Binchuan County, Yunnan Province, using the standard single mode fiber provided by China Mobile and the air gun source signal. The artificial drop weight signal was observed, which successfully verified the possibility of urban communication optical cable as for earthquake early warning and underground structure observation, and provided a new direction for DAS research and earthquake monitoring research [79].

In 2021, Hudson et al. proposed a method using a two-dimensional DAS array as an effective multi-component sensor to accurately characterize the transverse wave splitting caused by anisotropic ice structures. They used the glacial environment as an analogy to other earthquake environments, and the methodology and conclusions obtained in this work contributed to the implementation of DAS systems for applications in other microearthquake environments. When the DAS system was at a lower and near-quasi-static frequency, the spectral SNR and bandwidth measured by the superposition of multiple DAS channels were significantly improved compared to those of a single geophone [80] (Figure 22).

In the field of sensing, although the exploration of natural disasters such as earthquakes and tsunamis, or of unknown terrains such as sea beds and rift valleys, has continued, the accompanying risk factors must be taken seriously. At present, the distributed optical fiber acoustic sensing techniques can be very helpful to avoid danger. However, due to the weak RBS light, the optical fiber sensing signal decays exponentially, and it is difficult to achieve long-distance detection.

In 2021, Avinash et al. used a dark-fiber DAS array located in the Sacramento Basin of Northern California to detect small earthquakes in the geyser geothermal field at a distance of about 100 km [81] (Figure 23). All earthquakes of M ≥ 2.4 during the experiment were successfully detected by analyzing DAS data for 45 consecutive days. The latest progress in earthquake monitoring is summarized in Table 8.

### 4.3. Energy Exploration

Oil, natural gas, and coal are important strategic resources in China, and the exploration techniques for these resources have been continually studied. Energy exploration techniques can greatly reduce the cost of extraction, improve the accuracy of extraction, and improve the energy pattern in China [82]. Conventional exploration techniques consume large amounts of human and material resources, and are limited by high temperature and pressure, which prevent the exploration requirements from being met. By comparison, the DAS system with optical fiber cable as the main transmission body has a high spatial and temporal resolution, in addition to a large sensing distance, which can cope with the complex geological environment and meet the technical requirements of the surface detector and in-well detector. An increasing number of researchers have started to study the earthquake wave detection techniques of DAS [83].

To investigate the relationship between behavior and ground movement deformation during coal mining, in 2019, Chai et al. used distributed optical fiber monitoring techniques to record the strain on the ground surface [84]. The experiment proved that the distributed optical fiber monitoring technique was expected to replace traditional coal mine monitoring and provided a theoretical basis for surface subsidence prediction, geohazard evaluation, and surface subsidence control in mining areas.

In 2021, Wang et al., from the Laser Institute of Shandong Academy of Sciences, designed a distributed optical fiber acoustic monitoring system for oil and gas seismic wave exploration and development. The optical cable was used as a sensor to detect the sound signal, and the phase modulation and demodulation techniques based on back Rayleigh scattering were adopted to realize the test of 10 m spatial resolution and −145.35 dB sound pressure sensitivity. The field exploration of seismic bombs and guns was carried out [85] (Figure 24). The earthquake wave signal acquisition and processing were completed, and clear formation inversion information was obtained.

In 2021, Wamriew et al. proposed a new deep learning method for real-time/semi-real-time processing of large volumes of DAS data [86] (Figure 25). The method was trained on publicly available data from Phase 2C hydraulic fracturing augmentation at the FORGE research site near Milford, Utah, USA, and ray tracing was used in generating the training dataset. Finally, in situ DAS microearthquake data acquired from hydraulic fracturing operations were used for validation. The results showed that the model was able to learn the relationship between microearthquake waveform data and event location, and was a high accuracy velocity model. The latest progress for energy exploration is summarized in Table 9.

### 4.4. Underwater Positioning

Underwater positioning techniques comprise one of the important research elements in marine resource exploration and marine military defense. The DAS system has the ability to adapt to complex environments and has a wide monitoring range [87]. It can achieve long-distance underwater monitoring through submarine optical fiber cable and plays an important role in marine oil and gas mineral development, submarine optical fiber cable pipeline laying, maintenance, and other projects, and ship and submarine mobilization, in addition to marine catastrophic geological research and underwater archaeological exploration.

Submarine optical cable is expensive, and its environment is harsh. Determining how to monitor submarine optical cables is particularly important. In order to realize timely monitoring, and advance prediction of failure of, submarine optical cables, in 2021, Zhang et al. proposed a submarine optical cable detection system based on enhanced coherent optical time domain reflectometry (E-COTDR) [88]. In the experiment, the system achieved 121 km full coverage monitoring for multi-span cascaded submarine cables of more than 1000 km, and, at the same time, also measured the loss of submarine cables. The approach improved performance compared to that of traditional submarine cable monitoring methods (Figure 26).

In 2021, Rivet et al. conducted an experiment on a 41.5 km long optical cable near the French port of Toulon, using an oil tanker sailing near the optical cable and the position map of the detected optical cable. From the 5.8 km offshore water depth of 85 m to the 20 km offshore water depth of 2000 m, the acoustic signal measured by DAS was used for analysis, and beamforming was used to obtain the hull trajectory at the 85 m water depth [89] (Figure 27). At the 2000 m water depth, due to serious signal attenuation, the hull track was obtained, but the frequency band signal was still detected below 50 Hz.

In 2021, Liu et al. proposed an underwater localization system that effectively recovered acoustic signals [90]. The system was based on a phase-sensitive optical time domain reflectometer with 3D printed sensing elements as the base optical path, and used a time-difference (TDOA) algorithm for the 3D position. The method was flexible in its operation and could be changed to suit practical needs, and showed great potential for development.

In 2022, Xu et al. used an optical frequency comb (OFC) formed by multi-frequency detection pulses for underwater localization. The approach proved to be well-adapted for the underwater environment and provided a new measurement method for future underwater positioning [91]. The latest progress for underwater positioning research is summarized in Table 10.

### 4.5. Railway Monitoring

As a result of the global increase in traffic demand, railways are playing a more important role, and railway safety issues are becoming increasingly prominent. Train positioning and trajectory monitoring, track safety detection, and safety detection along the line are of great significance to the safe operation of railways [92]. Because railway tracks are exposed to nature all year round, they are subjected to wind, rain, freeze–thaw cycles, and train loads [93], which may lead to many unexpected situations. At present, railway inspection mainly relies on manpower and safety inspections of trains [94]. However, due to the characteristics of railway’s day and night operations, conventional railway monitoring methods have been unable to meet the growing demand. Thus, distributed fiber acoustic sensing techniques have become a key trend in railway inspection because they can achieve real-time detection of track conditions.

In 2021, Wang et al. took the high-speed railway track as the research object, constructed a track and train detection system based on distributed fiber acoustic sensing, and proposed a new track state detection scheme with the deep convolutional network as the core. In this system, the incident checks included those of a crack, beam joint, switch, and lower road. The final recognition accuracy rate reached 98.04% [95] (Figure 28).

Accurate tracking of the true position of trains on the track is the basis of all modern railway monitoring concepts. It is important to provide sufficient safe separation between trains at all times [96,97]. An accurate, reliable, and simple train tracking techniques is an essential foundation for these new concepts.

In 2019, Kowarik et al. analyzed data from Deutsche Bahn’s ICE 4 trains to locate train signals along with temporal or spatial directions in the data, using track-view, train-view, and bogie cluster data analysis. The approach allowed train speeds to be determined in three different ways, and the study presented new approaches for train monitoring [98]. In 2020, Christoph et al. proposed a real-time train tracking algorithm. The performance was tested in tunnels with standard cable trenches and on open tracks with directly connected cables [99] (Figure 29). The study provided a new idea for train positioning.

Artificial intelligence (AI) and machine learning (ML) are also very effective for train monitoring. In 2022, Huang et al. used optical fiber cable as a sensing and transmission tool to implement traffic monitoring and cable failure prevention on telecommunication networks with the help of artificial intelligence (AI) and machine learning (ML) technologies. Thus, the study provided strong support for the construction of future smart cities [100]. The latest progress for railway monitoring is summarized in Table 11.

## 5. Conclusions

This paper systematically reviews the research and application progress of DAS techniques. The latest research progress is specified in terms of polarization fading, coherent fading, spatial resolution, frequency response, signal-to-noise ratio, and detection distance. DAS techniques now also play a non-negligible role in applications such as soil salinity and ocean measurement. At present, DAS techniques are not mature enough and the event recognition rate for practical applications is low. There is still a considerable gap compared to conventional point sensors in terms of sensitivity and other aspects. A balance is needed in the relationship between pulse width, SNR, and sensing distance, and the relationship between frequency range and sensing range. With the breakthroughs in detection distance, sensitivity, multi-parameter monitoring, and multi-dimension monitoring, in addition to the combination with deep learning and neural networks, DAS techniques will play an important role in many fields due to their unique advantages.

## Figures and Tables

**Figure 1 sensors-22-06060-f001:**
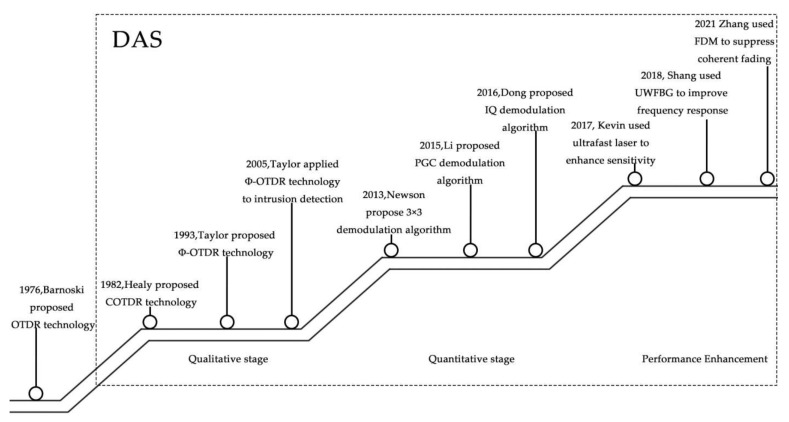
The history of DAS development.

**Figure 2 sensors-22-06060-f002:**
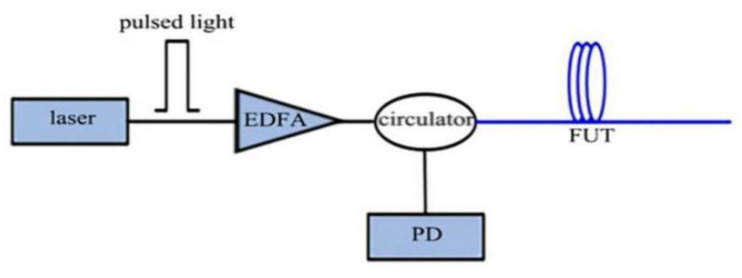
OTDR structure diagram.

**Figure 3 sensors-22-06060-f003:**
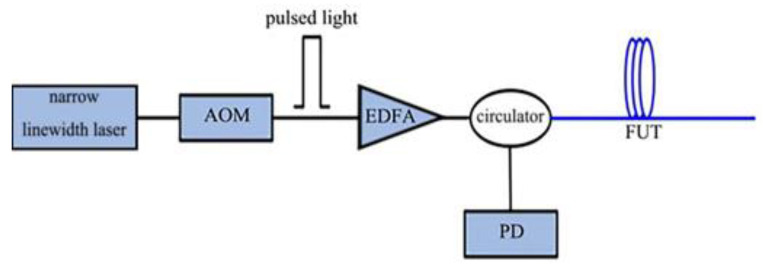
Φ-OTDR structure diagram.

**Figure 4 sensors-22-06060-f004:**
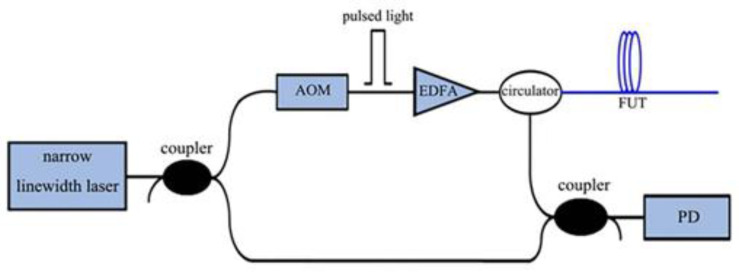
COTDR structure diagram.

**Figure 5 sensors-22-06060-f005:**
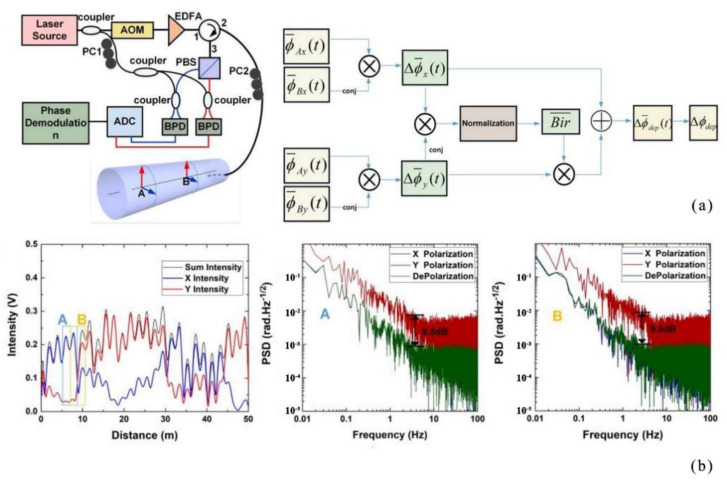
(**a**) Experimental setup and demodulation procedure. (**b**) Backscattered light intensity of backscattering enhanced fiber in different polarized states; PSD of fiber section A; PSD of fiber section B in X polarized state, Y polarized state and depolarized algorithm [33].

**Figure 6 sensors-22-06060-f006:**
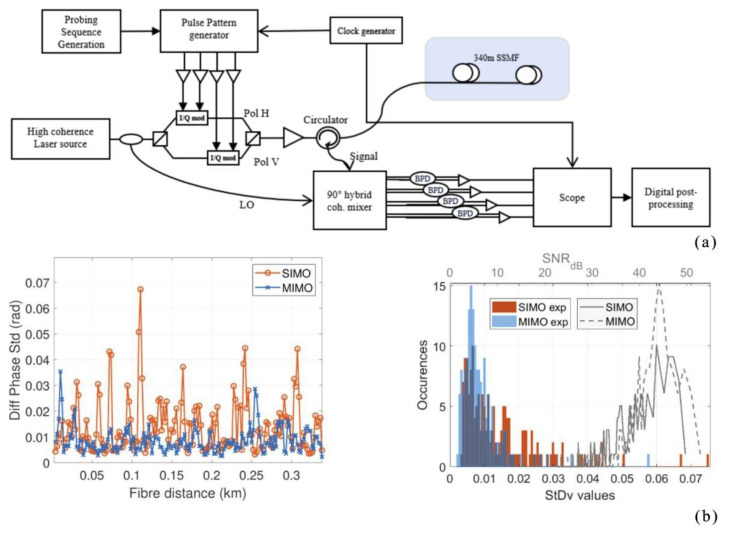
(**a**) Experimental setup. (**b**) SIMO and MIMO measurements on 340 m SSMF, no perturbation applied [35].

**Figure 7 sensors-22-06060-f007:**
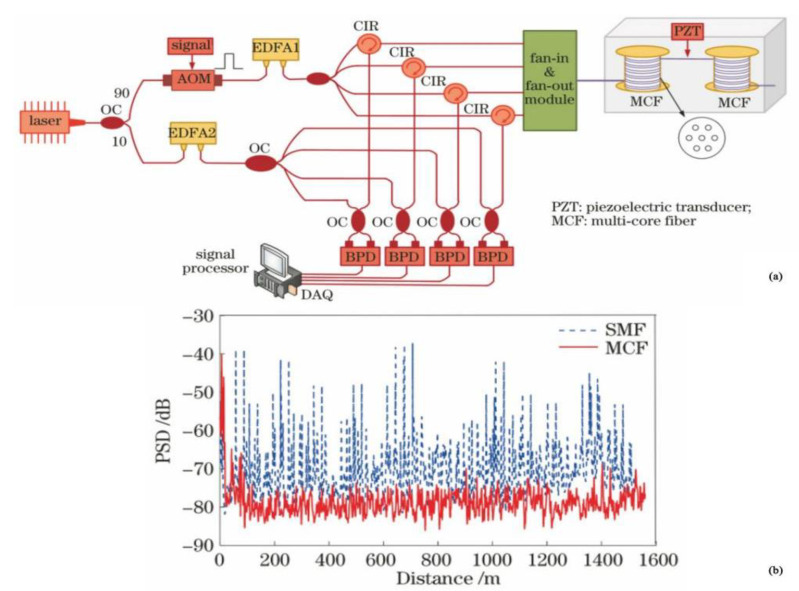
(**a**) Experimental setup. (**b**) PSD of SMF and MCF at the frequency of 2.5 kHz [36].

**Figure 8 sensors-22-06060-f008:**
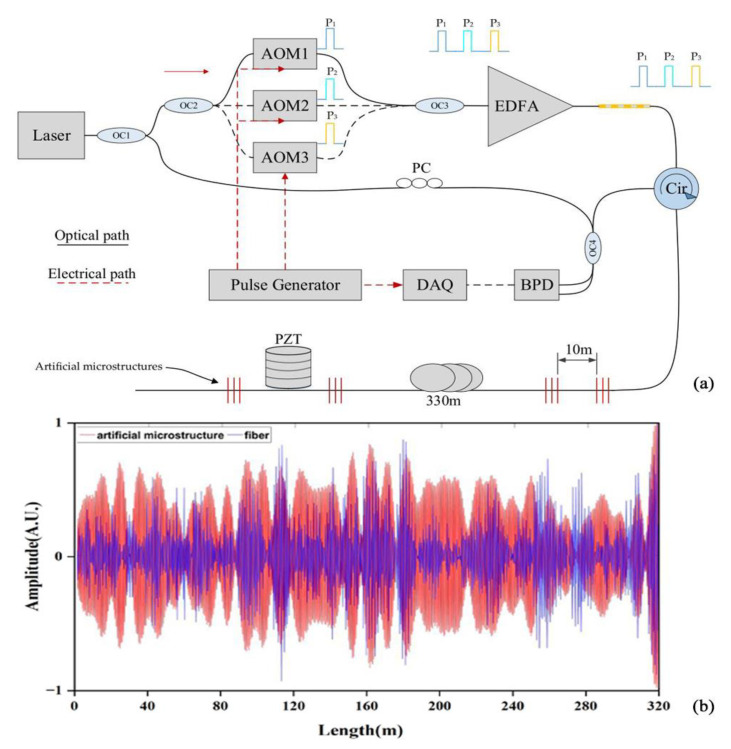
(**a**) Experimental setup. (**b**) The beat frequency signal [44].

**Figure 9 sensors-22-06060-f009:**
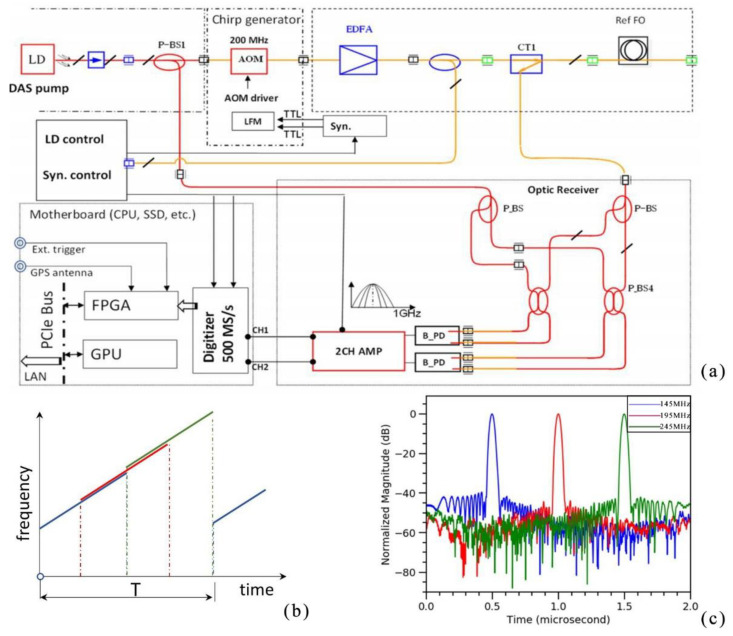
(**a**) Experimental setup. (**b**) Chirped pulse and sub-division into bands (offset for visibility, frequency change is linear over entire range). (**c**) Corresponding frequency bands used during signal processing [46].

**Figure 10 sensors-22-06060-f010:**
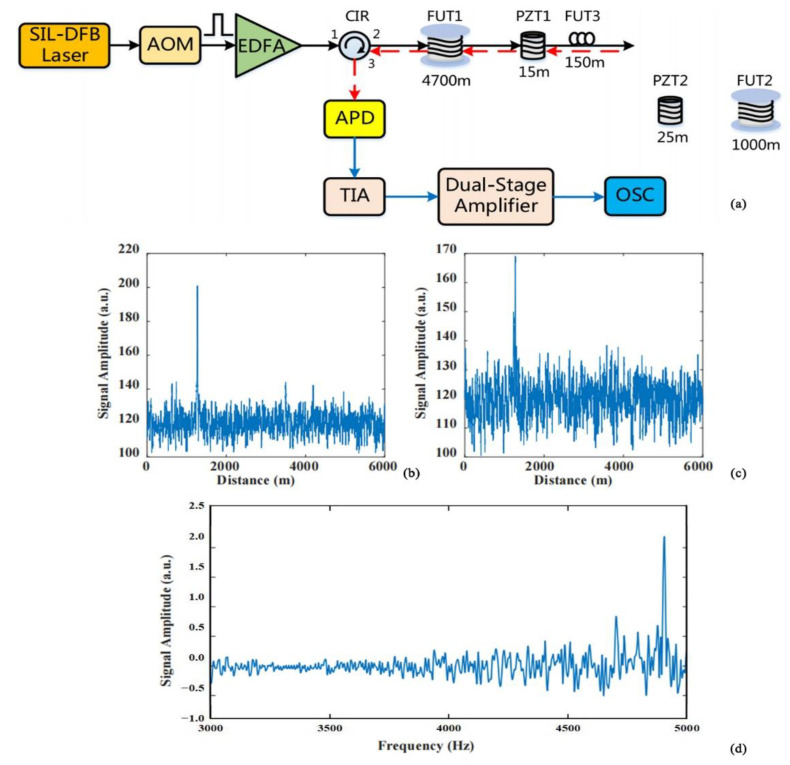
(**a**) Experimental setup. Localization of vibration sources at different frequencies: (**b**) 8 Hz; (**c**) 4.9 kHz. (**d**) FFT frequency spectrum at the 4.9-kHz vibration point [49].

**Figure 11 sensors-22-06060-f011:**
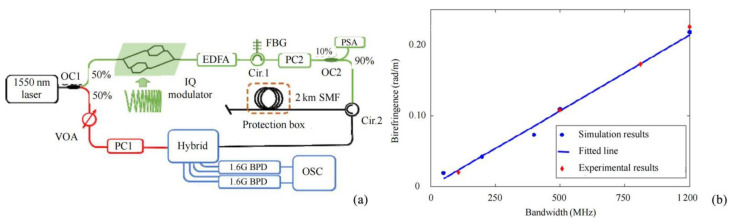
(**a**) Experimental setup. (**b**) Demodulated birefringence of simulation and experiments [51].

**Figure 12 sensors-22-06060-f012:**
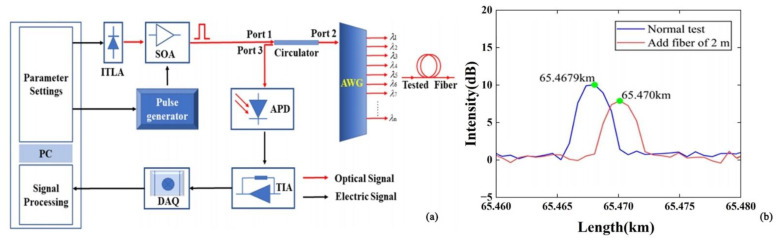
(**a**) Experimental setup. (**b**) The result of spatial resolution measurement [54].

**Figure 13 sensors-22-06060-f013:**
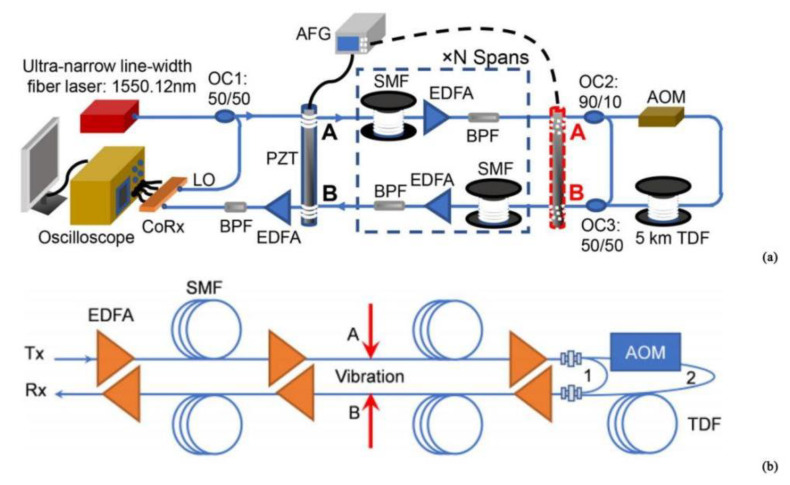
(**a**) Experimental setup. (**b**) Schematic diagram of the localization principle [57].

**Figure 14 sensors-22-06060-f014:**
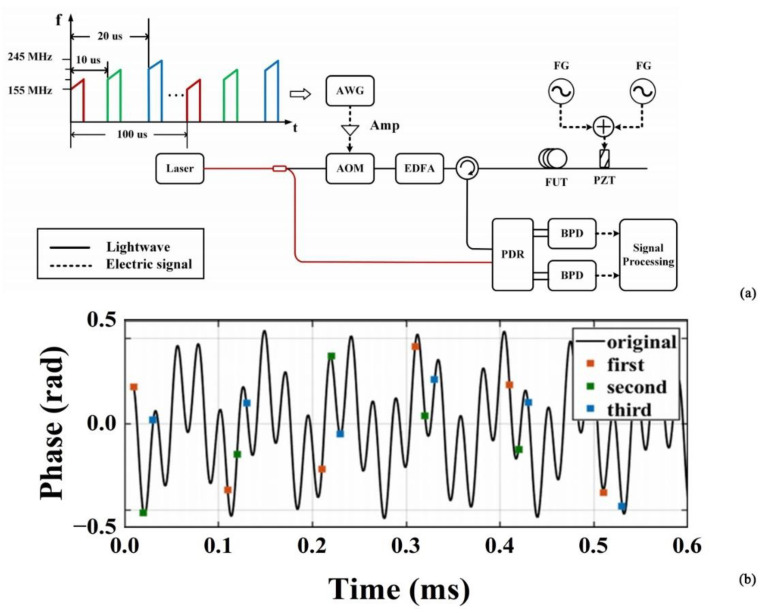
(**a**) Experimental setup; red is polarization-maintaining fiber. (**b**) Time domain recovered waveform of vibrations [60].

**Figure 15 sensors-22-06060-f015:**
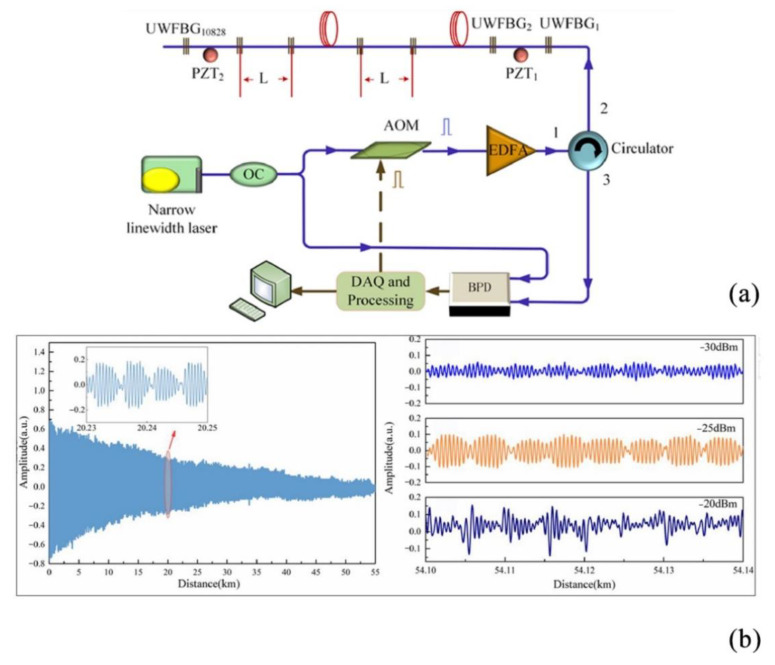
(**a**) Experimental setup. (**b**) Raw beat frequency signal [64].

**Figure 16 sensors-22-06060-f016:**
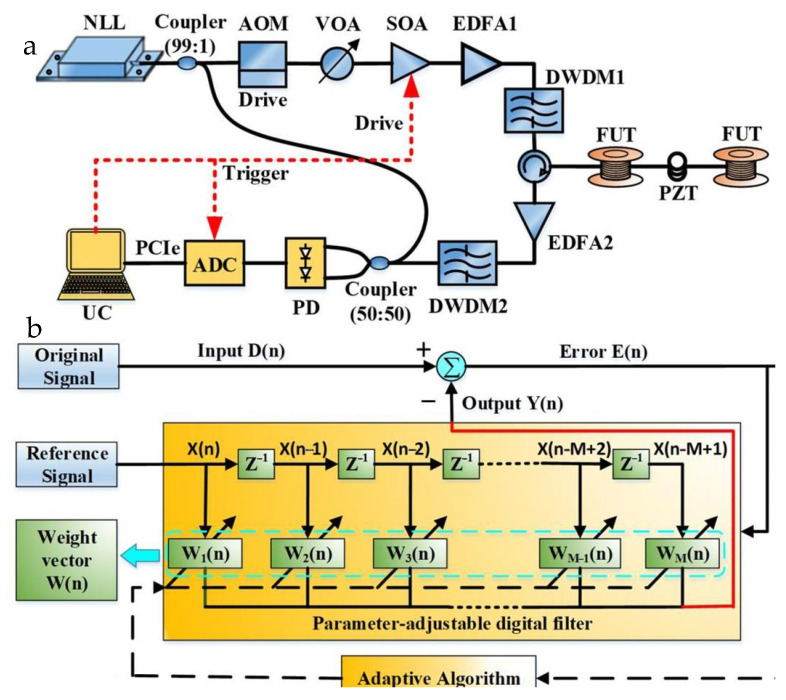
(**a**) Experimental setup. (**b**) Schematic diagram of adaptive filtering based on least mean square error (LMS) algorithm [66].

**Figure 17 sensors-22-06060-f017:**
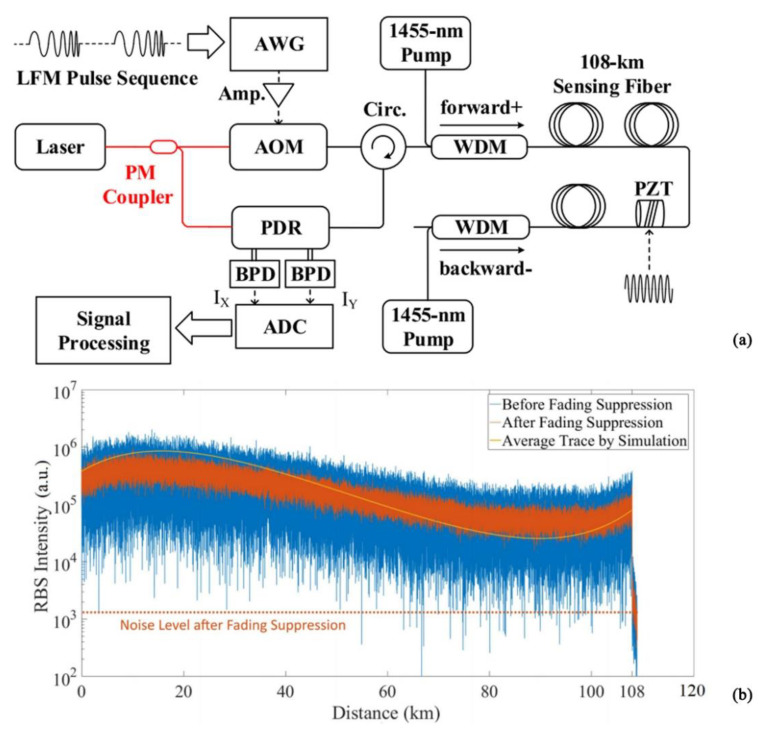
(**a**) Experimental setup. (**b**) RBS intensity–distance trace of 108 km sensing fiber with bi-directional first-order distributed Raman amplification [69].

**Figure 18 sensors-22-06060-f018:**
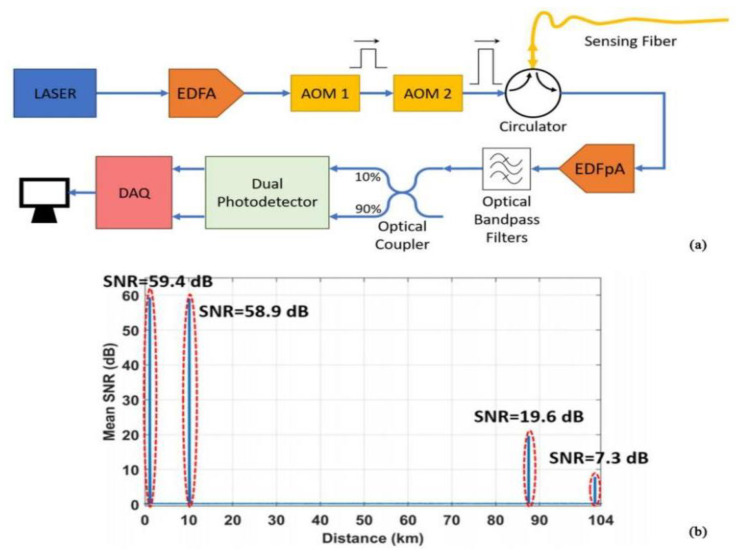
(**a**) Experimental setup. (**b**) Mean SNR versus distance along the test fiber [72].

**Figure 19 sensors-22-06060-f019:**
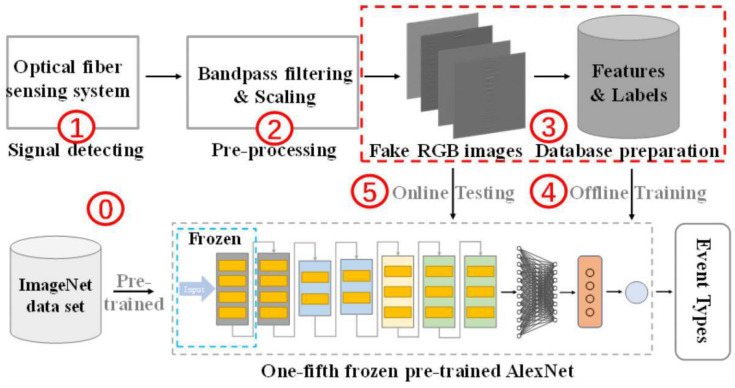
Classification method steps [74].

**Figure 20 sensors-22-06060-f020:**
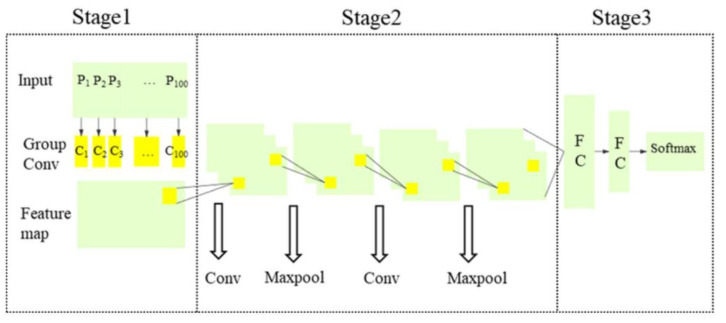
100 G-Net structure diagram [75].

**Figure 21 sensors-22-06060-f021:**
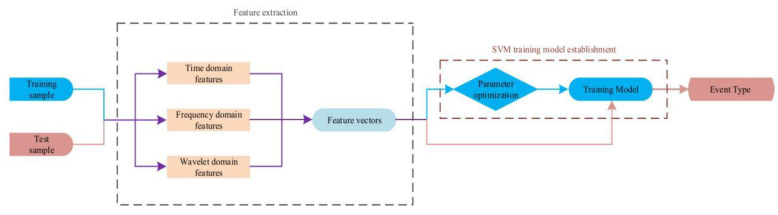
The classification algorithm flow [76].

**Figure 22 sensors-22-06060-f022:**
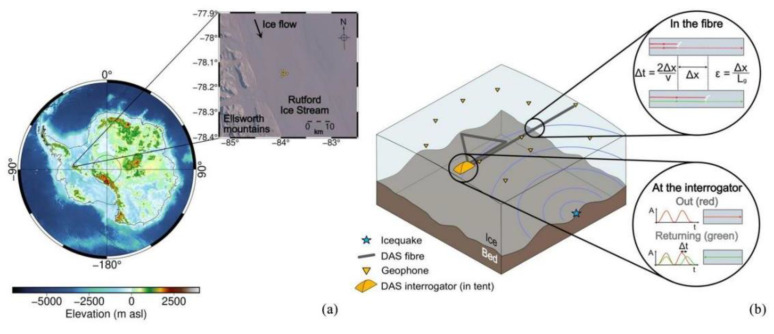
(**a**) Map and schematic showing experimental setup. (**b**) Schematic diagram of the experiment with the triangle and line fiber configurations [80].

**Figure 23 sensors-22-06060-f023:**
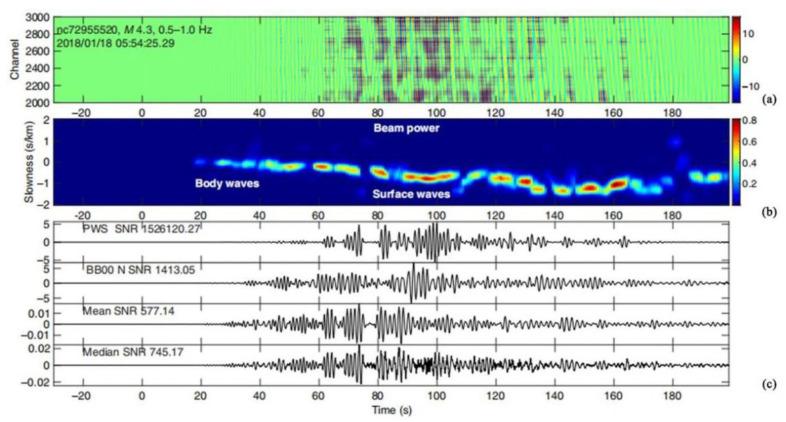
Beamforming results at channels 2000–3000 (subarray aperture 2 km) for an M4.3 earthquake. (**a**) Record section showing filtered waveforms. (**b**) Vespagram (beam power in the 0–1 range as a function of slowness and time). (**c**) Traces from the top to bottom [81].

**Figure 24 sensors-22-06060-f024:**
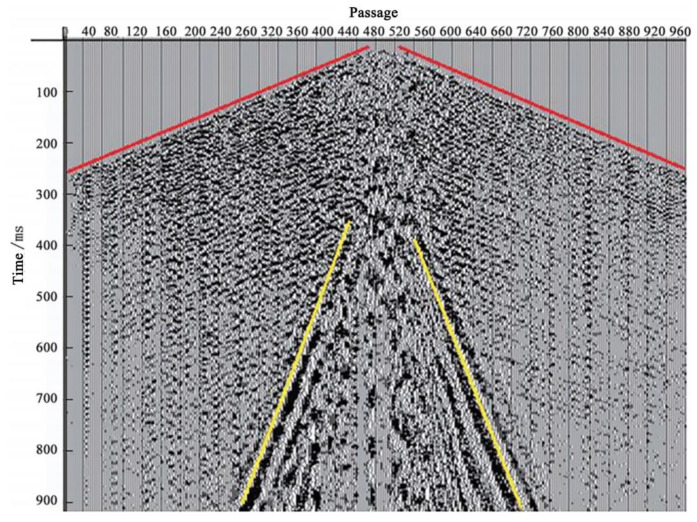
DAS earthquake wave test data [85].

**Figure 25 sensors-22-06060-f025:**
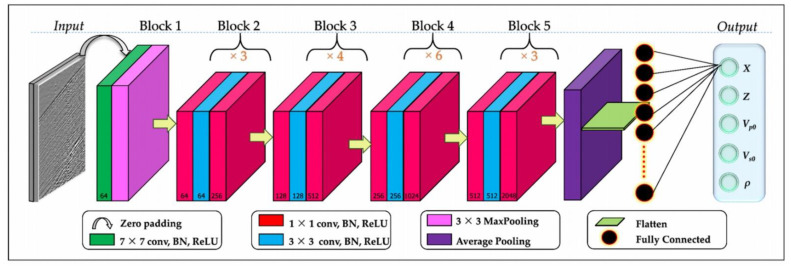
Deep convolutional neural network architecture used in the study. Green, blue, and red cuboids represent multi-channel feature maps with the number of channels shown at the bottom of the cuboids [86].

**Figure 26 sensors-22-06060-f026:**
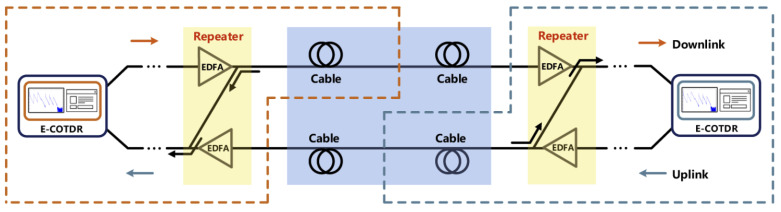
Schematic diagram of submarine cable link [88].

**Figure 27 sensors-22-06060-f027:**
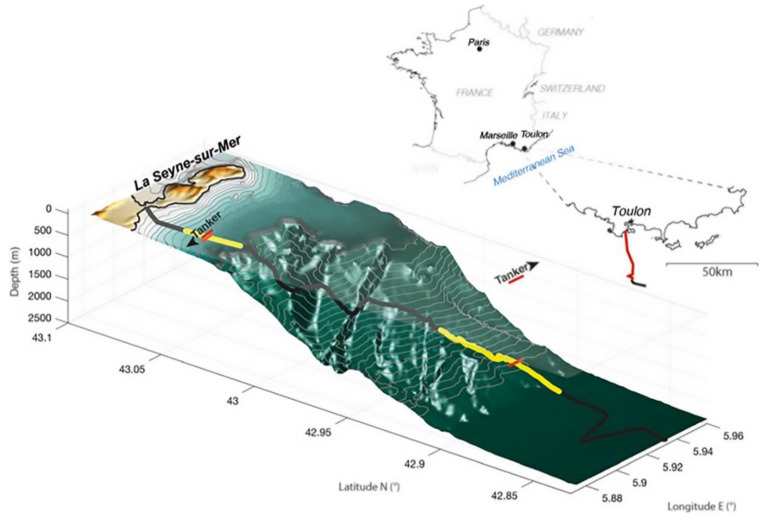
Position map of optical cable detection [89].

**Figure 28 sensors-22-06060-f028:**
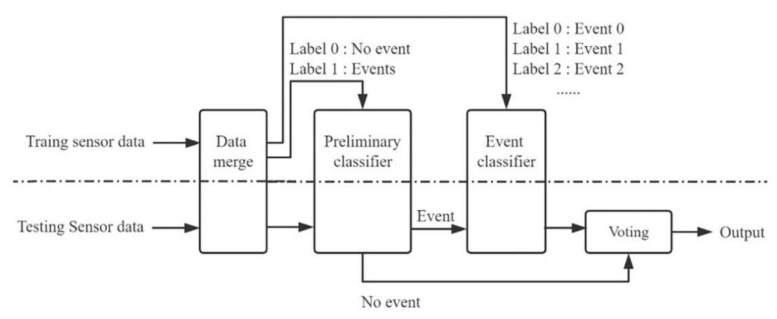
The diagram of the event recognition system [95].

**Figure 29 sensors-22-06060-f029:**
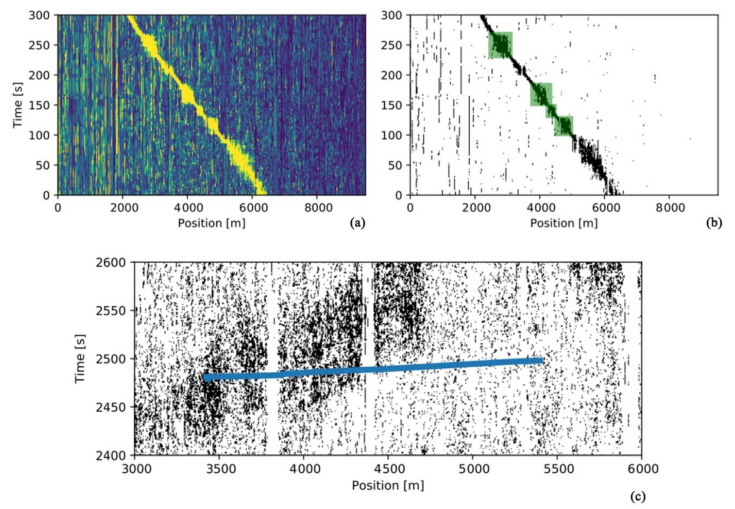
(**a**) Subsampled signal through band energy calculation. (**b**) Train detection result for a train on test site 2; the green boxes indicate coupled cable segments; the background without the train shows increased noise. (**c**) False positive train track due to false positive vibration detection [99].

**Table 1 sensors-22-06060-t001:** Research progress for suppression of polarization fading.

Published Date	Researchers	Polarization Fading Suppression Scheme	Performance
CJSI, 2014	Wu, et al.	Coherent and polarization maintaining light path structure	Interference fringe visibility up to 40%
LP, 2015	Alekseev, et al.	Dual-pulse diverse frequency probe signal	
OE, 2017	He, et al.	Phase-detection	SNR: 26 dB
CLEO, 2020	Sun, et al.	Dynamic birefringence estimation	Suppress about 9.5 dB noise
JLT, 2020	Rao, et al.	Bipolar Golay coding	Suppress about 7.1 dB noise
OE, 2020	Guerrier, et al.	Coherent-MIMO sensing	Improve sensitivity
AOS, 2021	Cai, et al.	Spatial diversity	Suppress about 5.2 dB noise
SR, 2021	Ogden, et al.	Frequency multiplexed pulse sequence	Strain noise: 0.6 pε/√Hz

**Table 2 sensors-22-06060-t002:** Research progress for suppression of coherent fading.

Published Date	Researchers	Coherent Fading Suppression Scheme	Performance
IEEE, 2018	Cai, et al.	DPSP	Sensing distance: 2.4 km
Elec, 2019	Zhang, et al.	FDM	Distortion rate: 1.26%
JLT, 2021	He, et al.	Phase-shift transform	Standard deviation of differential phase: 0.0224
Sens, 2021	Zhang, et al.	SDM	distortion rate: <2%
Sens, 2021	He, et al.	TGD-OFDR	Sensing distance: 80 km

**Table 3 sensors-22-06060-t003:** Research progress on for spatial resolution enhancement.

Published Date	Researchers	Spatial Resolution Enhancement Scheme	Spatial Resolution
ILE, 2016	Shang, et al.	Phase carrier demodulation algorithm	10 m
OE, 2019	He, et al.	Chirped pulse	2 m
Sens, 2019	Zhang, et al.	MSR	
OE, 2021	Ma, et al.	LiNbO_3_ straight-through waveguide phase modulator	10 m
AO, 2021	Zhu, et al.	DFB with OWRR	13 m
PS, 2021	Wang, et al.	Pulse-Compression	0.086 m
OL, 2021	Qian, et al.	CPCA	4 m
IEEE, 2021	Gong, et al.	DWDM-PON	2 m

**Table 4 sensors-22-06060-t004:** Research progress for frequency response enhancement.

Published Date	Researchers	Frequency Response Enhancement Scheme	Frequency Response
IT, 2016	Li, et al.	DBR fiber laser sensor	
IEEE, 2018	Shang, et al.	Broadband weak FBG array	1200 Hz @ 400 m
OFT, 2019	Zhang, et al.	UWFBG with FDM	440,000 Hz @ 330 m
IEEE, 2021	Yan, et al.	Ultra-long Distributed sensor	20,000 Hz @ 615,000 m
AO, 2021	Liu, et al.	FDM	2000 Hz @ 70,000 m
IEEE, 2021	He, et al.	Time delay sampling with FDM	47,000 Hz @ 10,000 m
OSA, 2021	Murrey, et al.	High-speed camera with time-gated local oscillator	400 Hz @ 2000 m

**Table 5 sensors-22-06060-t005:** Research progress on SNR enhancement.

Published Date	Researchers	Signal-to-Noise Ratio Enhancement Scheme	Signal-to-Noise Ratio
IEEE, 2017	Zhang, et al.	UWFBGs	58 dB
OSA, 2020	Yang, et al.	UWFBGs	59.2 dB
Sens, 2020	Jin, et al.	Least mean square error algorithm	42.2 dB
JLT, 2021	Cai, et al.	Dense multichannel signal integration	20 dB
COL, 2021	Zhang, et al.	Optimal peak-seeking and machine learning	
OSA, 2022	Yang, et al.	UWFBG array with coherent detection	40.01 dB

**Table 6 sensors-22-06060-t006:** Research progress for detection distance enhancement.

Published Date	Researchers	Detection Distance Enhancement Scheme	Detection Distance
Sens, 2018	Fu, et al.	BOTDR + Φ-OTDR	150.62 km
JLT, 2019	He, et al.	Bi-directional distributed Raman amplification	108 km
IEEE, 2019	Cedilnik, et al.	Two cascaded acousto-optic modulators	102.7 km
IEEE, 2019	Uyar, et al.	Low-loss optical fiber	125 km
OL, 2021	Masoudi, et al.	Low-loss enhanced- backscattering fiber	150 km

**Table 7 sensors-22-06060-t007:** Summary of perimeter security applications.

Published Date	Researchers	Methods
JLT, 2021	Shi, et al.	Transfer training recognition algorithm
IEEE, 2021	Shi, et al.	Security system with multi-domain feature fusion
IEEE, 2022	Ni, et al.	100 G-Net recognition algorithm

**Table 8 sensors-22-06060-t008:** Summary of earthquake monitoring applications.

Published Date	Researchers	Methods
SCP, 2019	Wang, et al.	Perimeter security
JSE, 2021	Hudson, et al.	Two-dimensional DAS array
SRL, 2021	Avinash, et al.	Dark-fiber DAS array

**Table 9 sensors-22-06060-t009:** Summary of energy exploration applications.

Published Date	Researchers	Methods
OFT, 2019	Chai, et al.	Perimeter security
SDS, 2021	Wang, et al.	Propose monitoring system
Sens, 2021	Wamriew, et al.	Deep learning methods for real-time/semi-real-time data processing

**Table 10 sensors-22-06060-t010:** Summary of underwater positioning applications.

Published Date	Researchers	Methods
JASA, 2021	Rivet, et al.	Detection of oil tankers at sea
OE, 2021	Liu, et al.	Underwater localization system
OE, 2021	Zhang, et al.	Submarine cable
OC, 2022	Xu, et al.	OFC

**Table 11 sensors-22-06060-t011:** Summary of railway monitoring applications.

Published Date	Researchers	Methods
Sens, 2019	Kowarik, et al.	Cluster data analysis
OE, 2020	Christoph, et al.	Real-time train tracking algorithm
OC, 2021	Wang, et al.	Track train detection system
SPIE, 2022	Huang, et al.	AI and ML technologies

## Data Availability

Data openly available in a public repository.

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
