# Peer review of "Research Progress in Distributed Acoustic Sensing Techniques"

_sensors, 2022, doi:10.3390/s22166060_

Round 1

Reviewer 1 Report

The authors try to describe the progress in DAS technology, such as principles, technical difficulties, and corresponding solutions. Unfortunately, I did not see too much useful information in this review manuscript. I do not recommend publishing this manuscript in Sensor.

Reviewer 2 Report

This paper reviewed the latest developments of phase-sensitive optical time-domain reflectometry (phase-OTDR). The review doesn’t cover the latest trends in phase-OTDR progress. In recent years, there are many review papers in literature based on phase-OTDR systems. I don’t see clear operating principles, any rigorous analysis, and a summary. The equations presented in the manuscript don’t give any clear view of the operating principles and also need proper references. Overall, I don’t feel this manuscript includes enough phase-OTDR/DAS progress to be published in Sensors.  

The authors could significantly improve their manuscript by including, (i) AI/ML based phase-OTDR/DAS, (ii) dual (and >2) frequency probe pulses, (iii) specialty optical fibers/Rayleigh enhanced optic fibers, (iii) extended applications with a figure, which shows potential applications. A summary table for each section shows recent developments with their specifications. 

Reviewer 3 Report

Das-distributed optical fiber acoustic sensing technology is based on phase-sensitive optical time-domain reflectometry, which is a new sensing technology to realize continuous distributed detection of acoustic signals by using Rayleigh backscatter effect of optical fiber. Article system combed the distributed optical fiber sensing technology in the research process, the physical mechanism and technical difficulties and solutions, pointed out the direction of future development of the technology of DAS, the article selected topic, is of great significance to the technological development in this field, but there are some problems need to be solved before published article, is listed as follows.

1. It is suggested to give a chart of the development history and typical application of DAS (it is suggested to draw a chart of the development history of DAS); Chronology - Milestones

2. In FIG. 5 line168, there is an error in the annotation of the referenced article map, for example, the labels of (a) and (b) are repeated. Please correct the reference to consider whether to apply for copyright. Please refer to it properly. Please carefully check whether there are similar mistakes in other parts and correct them. In addition, the referenced figures are not very clear, such as Figure 7 (b), Figure 16, Figure 17 and Figure 21

3. The article did not give further comments after listing typical cases, and suggested supplementation, so as to clarify the development context for readers. For example, in 3.1, recommendation line185, it is suggested to give a comprehensive review and summary. Similar include 3.2,3.3, etc., please supplement and modify.

4. The article gives a general description of the case, but lacks in-depth summary and necessary comments, so many parts of the research content still need to be further deepened. As a leading review in this field, it needs to be of a certain height and forward-looking, and it is hoped that the author can read through the article to make necessary modifications and improvements.

Reviewer 4 Report

This paper gives a review of the DAS technology, which could be interesting to readers. This paper could be considered to be published if the follow problems could be fixed:

1.     The logical thinking of this paper is not very clear, and the connection between contexts is not close. The title of the paper is about the research progress of DAS. The author mentioned COTDR many times in the paper, but did not explain the relationship between COTDR and DAS; For example, the three sections of the principle part are independent of each other and lack of cohesion.

2.     In the introduction, it is not clear what the author’s main point is. The author divides the development of DAS into three eras (line 43-59), but the contents expressed in the three paragraphs are not quite consistent. And I don’t think this division is accurate. Normally, we divide phi-OTDR into DVS and DAS, so DVS and DAS are peer rather than inclusive.

3.     The paper is more of a chronological list of some research progress, lacks of summary and reflection on the research, and it is more like a report than an article.

4.     Please consider whether there is a problem with corner mark (a) and (b) of figure 5.

5.     Chapter 3.2 introduces in detail the methods currently used to improve spatial resolution. Should the spatial resolution to be set as high as possible? As far as we know, there is also a relationship between spatial resolution and SNR, so we suggest it should be supplemented.

6.     This paper analyzes the technical difficulties and solutions of DAS technology in terms of polarization fading, spatial resolution, frequency response, and long-distance detection. It is suggested to also include other technical difficulties, such as the suppression of coherent fading noise.

7.     It seems that the research progress of the author’s investigation is not yet comprehensive, for example, in the application of underwater monitoring, the following manuscripts use the enhanced COTDR system to realize the monitoring function of DAS, but I cannot see a reference to it:

X. Chen, N. Zou, L. Liang, R. He, J. Liu, Y. Zheng, F. Wang, X. Zhang and Y. Zhang, “A submarine cable monitoring system based on enhanced COTDR with simultaneous loss measurement and vibration monitoring ability,” Optics Express 29(9), 13115-13128 (2021).

Round 2

Reviewer 2 Report

The authors addressed all the comments. The tables, which describe the recent progress of polarization and coherent fading suppression, spatial resolution, and frequency response increment are of great interest to the readers. The manuscript has been improved and covers all the recent developments of phase-OTDR. I still suggest English language editing before final submission. 

Reviewer 3 Report

After the overhaul, the author responded positively to and modified my suggestions, and made necessary analysis and comparison on the literature review. The paper has been greatly improved in general, which plays a certain guiding role in summarizing and combing the knowledge in this field. Some languages need to be modified and refined, so I agree to publish the manuscript in the Journal of Sensors.

Reviewer 4 Report

The author responded well to the comments,therefore, I suggest the work is worthy to be published in Sensors.
